# Evaluating Cultural Adaptability of a Large Language Model via Simulation of Synthetic Personas

**Louis Kwok, Michal Bravansky, Lewis D. Griffin**

Department of Computer Science
University College London
66 - 72 Gower St, London, United Kingdom
{louis.kwok.22, michal.bravansky.22, l.griffin}@ucl.ac.uk

## Abstract

The success of Large Language Models (LLMs) in multicultural environments hinges on their ability to understand users' diverse cultural backgrounds. We measure this capability by having an LLM simulate human profiles representing various nationalities within the scope of a questionnaire-style psychological experiment. Specifically, we employ GPT-3.5 to reproduce reactions to persuasive news articles of 7,286 participants from 15 countries; comparing the results with a dataset of real participants sharing the same demographic traits. Our analysis shows that specifying a person's country of residence improves GPT-3.5's alignment with their responses. In contrast, using native language prompting introduces shifts that significantly reduce overall alignment, with some languages particularly impairing performance. These findings suggest that while direct nationality information enhances the model's cultural adaptability, native language cues do not reliably improve simulation fidelity and can detract from the model's effectiveness.

## 1   Introduction

Personalization of Large Language Models (Radford et al., 2019), such as GPT-4 (Achiam et al., 2023), Gemini (Team et al., 2023), and Llama (Touvron et al., 2023), is critical for their wider adoption (Chen et al., 2023). Equipping LLMs with the capability to learn and adapt to individual user profiles will allow these models to offer more relevant, context-aware, and personalized responses (Tan & Jiang, 2023; Tan et al., 2024).

One dimension of personalization is cultural awareness (Hershcovich et al., 2022). AI models capable of producing responses that respect the user's nationality and cultural background can aid in high-stakes communication, such as therapy chatbots (Wang et al., 2021), translation (Yao et al., 2023), creative writing (Shakeri et al., 2021), and human modeling (Argyle et al., 2023).

The predominant approach to ensuring LLMs have this cultural awareness is training on large corpora of multilingual data (Conneau et al., 2019; Guo et al., 2020; Singh et al., 2024). Such training, however, still produces models biased towards the English language and Anglo-centric culture (Talat et al., 2022; Havaldar et al., 2023), while other backgrounds are underrepresented or misrepresented (Hovy & Yang, 2021; Ahia et al., 2023; Shafayat et al., 2024; Mirza et al., 2024).

In this study, we explore these themes from the perspective of conveying nationality-related information. Moreover, we address two critical questions: Firstly, how effective are different methods in conveying a person's nationality to an LLM? Secondly, how accurately are these national traits depicted in different multilingual settings?

Rather than relying on problematic subjective cultural alignment (Taras et al., 2009), we adopt an approach inspired by synthetic human modeling (Argyle et al., 2023; Griffin et al., 2023). By revisiting a multi-national psychological experiment from Bos et al. (2020) and utilizing GPT-3.5 (Ouyang et al., 2022), we simulate a population of synthetic human profiles with diverse nationalities. We prompt the LLM with specific human profiles, with or without information related to their nationality, along with a news stimulus from the original study, and generate completions to simulate responses to a predefined questionnaire. The nationality context of the human profile is provided either by specifying the individual's country of residence or translating the prompt into their native language.

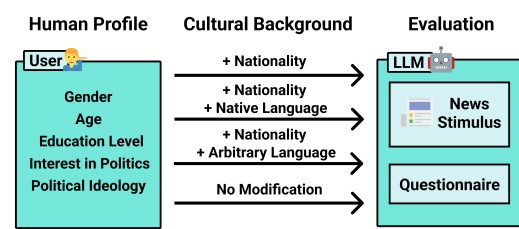

Figure 1: A specific human profile is defined, enriched with nationality or language, and evaluated against the ground-truth results from Bos et al. (2020).

We evaluate the effectiveness of different methods of conveying the nationality of simulated participants reflected by the accuracy of the model's responses, using the results of the original study as a benchmark. While we acknowledge the limitations of this methodology in fully capturing the nuances of cross-cultural AI, we believe this work can pave the way for creating more personalized and inclusive AI applications, and we encourage further research into the crucial area of culturally sensitive LLMs.

## 2 Related Work

### 2.1 Multilingual and Multi-Cultural Aspects of Large Language Models

Modern LLMs have gained multilingual capabilities through training on sufficiently large corpora including multilingual data (Brown et al., 2020). Despite LLMs' ability to transfer knowledge and competences across languages (Zhang et al., 2023), they still reflect the overarching distribution of the datasets (Kunchukuttan et al., 2021) and tend to demonstrate bias towards high-resource languages (Blasi et al., 2022). Nevertheless, they can be utilized to accurately classify sentiment, offensiveness, or moral foundations across a wide range of languages (Rathje et al., 2023; Přibáň et al., 2024).

LLMs are able to capture cultural differences, but this capability can also lead to the elicitation of biases and stereotypes. The frequency of gender bias produced by LLMs varies based on the language of the prompt (Zhao et al., 2024; Stańczak et al., 2023). Furthermore, these stereotypes can be elicited even by different English dialects (Hofmann et al., 2024).

There have also been studies evaluating the cultural understanding of LLMs through survey data (Arora et al., 2023). Durmus et al. (2023) showed that steering the model through prompting to consider a given country's perspective can enhance its ability to replicate the opinions of that population, but also elicits stereotypes. Cao et al. (2023b) observed that the effectiveness of this steering varies across different cultures.

As the concept of an emotion is strongly dependent on the language and associated culture (De Bruyne, 2023), their treatment by LLMs have been studied in multilingual and multi-cultural settings. Barreiß et al. (2024) observed that in a zero-shot setting, English-based prompts are better at classifying emotions than target-language ones. Additionally, Havaldar et al. (2023) showed that a prompt in a target language produces a less culturally-aware emotional response than communicating in English with a prefix stating the nationality.

### 2.2 Human-like Characteristics of Large Language Models

There has been considerable interest in studying the behavioral patterns of Large Language Models by treating them as participants in psychology experiments originally designed for humans (Hagendorff, 2023).

These studies have produced inconclusive results regarding opinion- and personality-based analyses of these models. For instance, Miotto et al. (2022) tested GPT-3 using questionnaire-based methods, only to find that the model's exhibited personality strongly varies with different sampling parameter settings. Santurkar et al. (2023) observed that the opinions expressed by LLMs are misaligned with those of any U.S. subgroup.

Despite these models possibly not exhibiting stable human-like characteristics of specific subgroups on their own, it is possible to adjust them to do so. Safdari et al. (2023); Hwang et al. (2023); Pan & Zeng (2023) have observed that LLMs are capable of simulating reliable and valid personality traits under specific prompting configurations. These characteristics are more accurate when produced by instruction-tuned models and can be shaped to simulate specific human profiles. Additionally, Abdulhai et al. (2023) showed that adversarially constructed prompts can control and make the LLMs elicit specific moral values.

In conjunction with this work, Jiang et al. (2023) define an "LLM persona" as an LLM-based agent prompted to generate content that reflects certain personality traits. By instructing them with the Big Five personality model, they show that their results on both the Big Five personality test and in a subsequent story-writing task are consistent with those of their defined personalities.

Apart from prompt-based techniques utilizing the model's in-context capability to simulate human characteristics, Mao et al. (2023) have used model-editing to control the model's opinions on specific topics. Additionally, Reinforcement Learning from Human Feedback (RLHF) is attributed as one of the possible causes of the pro-environmental, left-libertarian stance of chatbots like GPT-4 (Hartmann et al., 2023), as RLHF models are more likely to express liberal values, while the pre-trained LLMs are more associated with conservative perspectives (Safdari et al., 2023).

These overall results are consistent with the viewpoints of Shanahan et al. (2023). They argue that dialogue agents can be viewed as role-playing a number of characters at the same time which only start producing responses corresponding to a single individual when sufficient context is provided.

### 2.3 The Role of LLMs in Synthetic Behavior Simulation

This ability to mimic different human participants has propelled research into behavioral simulation that are enabled by these generative models (Mills et al., 2023; Dillion et al., 2023) with the benefit of relieving the cost of collecting human data (Kennedy et al., 2018; Keeter et al., 2017).

Argyle et al. (2023) propose "silicon sampling," which enables the generation of survey responses similar to those of humans using language models by prompting the LLM with a set of predefined individual-level characteristics about the population. To overcome the need to collect these characteristics before conducting an experiment, Sun et al. (2024) bases them on the demographic distribution of the population.

LLMs have been used to reproduce classic economic, psycholinguistic, and social psychology experiments (Aher et al., 2023). Additionally, they have been shown to mirror human behavior in politics (Wu et al., 2023), or to predict public opinion (Chu et al., 2023; Törnberg et al., 2023).

There are limitations to this simulation approach. Dominguez-Olmedo et al. (2023) observed that ordering and labeling biases affect LLMs' alignment with human behaviors. Leng & Yuan (2023) showed that LLMs demonstrate a pronounced fairness preference, and weaker positive reciprocity when compared to human results. Additionally, they might exaggerate effects that are present within humans due to reduced variance (Almeida et al., 2023).

## 3 Methodology

To address the questions posed in the introduction —"Firstly, how effective are different methods in conveying a person's nationality to an LLM?" and "How accurately are these na-

tional traits depicted in different multilingual settings?" — we performed three experiments all based on the study conducted by Bos et al. (2020), which assessed the degree to which human participants were persuaded and mobilized by populist news material. We have chosen to replicate this study over attitude-based ones such as the World Values Survey[1] and Pew Global Attitudes Survey[2] for four key reasons:

1. Large language models have been shown to overgeneralize on shared values of different nationalities, often resulting in stereotyping (Naous et al., 2023; Cao et al., 2023a). While this is an important area of study, we chose to minimize the effect of such biases by choosing an out-of-domain experimental setting.

2. To avoid assessing mere cultural knowledge of the LLM, we sought a setting that provides extensive subject-level information and their initial psychological state. This state is provided within the experiment by Bos et al. (2020) through relative deprivation ratings, which imply participants' initial susceptibility to persuasion.

3. The chosen study analyzes reactions to stimuli, effectively combining survey-style questionnaires with a classical psychological experimental setting. This approach enhances both the internal and external validity of the experiment (Atzmüller & Steiner, 2010).

4. There is growing concern about the potential dangers of LLMs' personalization capabilities in the field of persuasion (Griffin, 2023; Matz et al., 2024; Hackenburg & Margetts, 2024). This makes persuasion a relevant and important topic to investigate.

However, we recognize that the chosen study also has limitations, primarily due to its predominant focus on European nationals.

For each simulated participant, a news stimulus was provided and the LLM's response to a questionnaire was subsequently generated. Based on the recorded results, we analyze how the fidelity of the simulation's responses is affected by the methods used to specify the nationality of the simulated participants. Our first experiment analyzes the effect of explicitly stating the country of residence of simulated participants. Our second experiment investigates the effect of the language used when prompting the LLM. Our third experiment focuses on the impact of aligning the language of the prompt to that of the simulated participant.

### 3.1 Synthesizing Personas

In the original human study, participants filled out a questionnaire detailing their demographic information and self-rated relative deprivation, the latter being a measure of feelings of political, social, and economic vulnerability. They were then exposed to a news article forecasting adverse economic developments. 25% of participants were exposed to the factual story, 25% to a version which placed the blame on political elites, 25% to another version blaming immigrants, and the final 25% to a version placing blame on both groups. After reading the article, participants rated their agreement with five statements on a scale of 1-7: two statements assessing how persuaded they were, and three statements assessing how mobilized to action they were, by the article. The averages of these ratings were used to determine each participant's persuasion and mobilization scores respectively.

The questionnaire used in Bos et al. (2020)'s study was adapted to a format suitable for LLM simulation, illustrated in Figure 2. Using supplementary information from the original study, we were able to synthesize the same set of personas. The sample of 7286 simulated participants spanned 15 countries, comprising 14 European countries and Israel (see Table 3 of the Appendix). This yielded an average of 486 participants per country (s.d. = 74). As a compromise for the LLM simulation, the photo in the article (constant in all versions) was replaced with descriptive alt text.

---

[1]https://www.worldvaluessurvey.org/
[2]https://www.pewresearch.org/

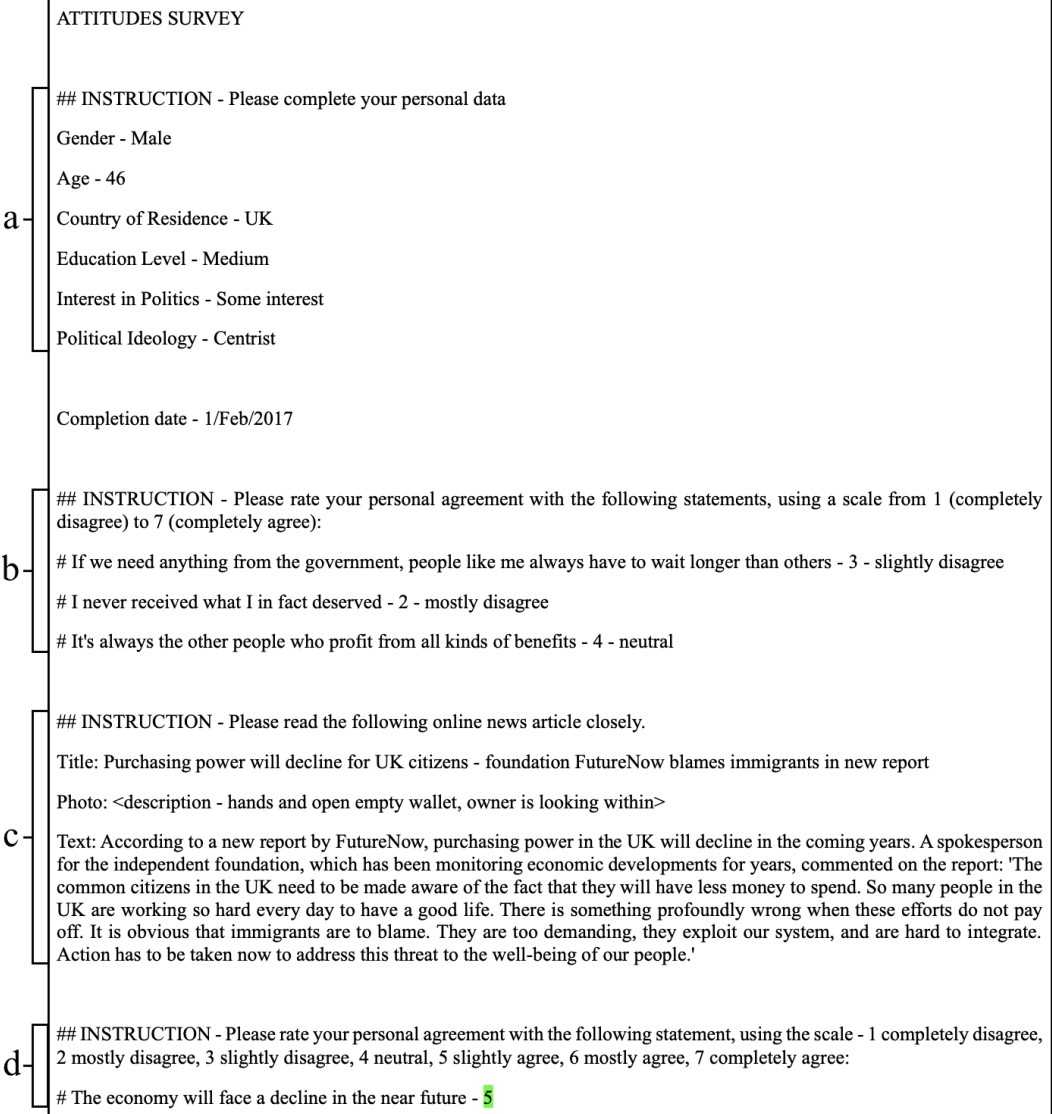

ATTITUDES SURVEY

**a)**
## INSTRUCTION - Please complete your personal data

Gender - Male

Age - 46

Country of Residence - UK

Education Level - Medium

Interest in Politics - Some interest

Political Ideology - Centrist

Completion date - 1/Feb/2017

**b)**
## INSTRUCTION - Please rate your personal agreement with the following statements, using a scale from 1 (completely disagree) to 7 (completely agree):

# If we need anything from the government, people like me always have to wait longer than others - 3 - slightly disagree

# I never received what I in fact deserved - 2 - mostly disagree

# It's always the other people who profit from all kinds of benefits - 4 - neutral

**c)**
## INSTRUCTION - Please read the following online news article closely.

Title: Purchasing power will decline for UK citizens - foundation FutureNow blames immigrants in new report

Photo: <description - hands and open empty wallet, owner is looking within>

Text: According to a new report by FutureNow, purchasing power in the UK will decline in the coming years. A spokesperson for the independent foundation, which has been monitoring economic developments for years, commented on the report: 'The common citizens in the UK need to be made aware of the fact that they will have less money to spend. So many people in the UK are working so hard every day to have a good life. There is something profoundly wrong when these efforts do not pay off. It is obvious that immigrants are to blame. They are too demanding, they exploit our system, and are hard to integrate. Action has to be taken now to address this threat to the well-being of our people.'

**d)**
## INSTRUCTION - Please rate your personal agreement with the following statement, using the scale - 1 completely disagree, 2 mostly disagree, 3 slightly disagree, 4 neutral, 5 slightly agree, 6 mostly agree, 7 completely agree:

# The economy will face a decline in the near future - 5

Figure 2: Format of a sample prompt used in the GPT-3.5 simulation. The prompt is intended to read like a semi-complete questionnaire, with the final numeric response (highlighted) provided by GPT-3.5. Key sections of the prompt are indicated by letters. a) Demographic information of the simulated participant. b) Relative deprivation ratings of the simulated participant in response to probe statements. c) The version of the news article shown to the simulated participant. In this example the anti-elite, anti-immigrant version is shown. d) The final instruction and a probe statement for GPT-3.5 to provide a single numerical response to.

### 3.2 Experiment 1: Effect of Indicating Nationality

The first experiment assessed the effect of indicating nationality of a simulated participant. The simulation was run twice: once with nationality information unmasked ('Country of Residence' in (a) and all mentions of the country in section (c) of the prompt as shown in Figure 2), and once with it masked. In both simulations, all prompts were in English, regardless of the nationality of the simulated participant.

### 3.3 Experiment 2: Effect of using a Single Language to Simulate Multinational Participants

In the second experiment, nationality information was present in all prompts, and the language of the prompt was varied. The simulation was run 12 times, with each run simulating in the same language regardless of the participants' nationalities. For example, in the first run, all subjects were prompted in English regardless of their nationality, in the second run, all subjects were prompted in French, and so on. The twelve languages used are the majority spoken languages in the 15 countries surveyed by Bos et al. (2020) as shown in Table 3 of the Appendix.

### 3.4 Experiment 3: Effect of using Native Languages to Simulate Multinational Participants

In the third experiment, nationality information was present in all prompts and the language of the prompt varied throughout the dataset. In the main condition, the prompting language matched each simulated participant's native language. In a *country-shuffled* condition, the prompting languages were randomly assigned to each country, while keeping the number of countries using each language constant. In a *full-shuffled* condition, the prompting languages were randomly assigned to each participant while maintaining the overall distribution of languages used. For both shuffling conditions, the simulation was performed 100 times to assess the variability of results.

### 3.5 Prompting Procedure

All experiments were conducted through the use of the OpenAI API to access the GPT-3.5-Turbo-1106 model. Five prompts were sent to GPT-3.5 for each simulated participant, where each prompt consisted of the pre-generated survey appended with one of the five probe statements that assessed either the participant's persuasion or mobilization after consuming populist material. To each prompt, the LLM returns a single number as its response. Following the original study, each participant's persuasion and mobilization scores were calculated by taking the average of the persuasion and mobilization ratings respectively provided by the model. Prompt translations in all languages were verified by native speakers who are also fluent in English to ensure that no linguistic meaning of the survey was lost or altered and no gendered biases were introduced. We have made the generation pipeline publicly available[3].

## 4 Evaluation

Similar to the analysis used in the original study, we perform regression analyses to fit the linear models shown in equation 1 and equation 2. In these models, the boolean features $E$ and $I$ indicate whether the participant was exposed to anti-elitist and anti-immigrant framing respectively, while the feature $1 \leq D \leq 7$ encodes the mean relative deprivation rating of the participant. $\bar{P}$ and $\bar{M}$ are the mean persuasion and mobilization ratings. The $C$ terms are country-specific, with the index $i$ selected to match the country of the participant. The regression analysis estimates values for the $C$ terms (so that the $P$ versions, and respectively the $M$ versions, have zero mean across countries) and the $\lambda$ coefficients. For both P and M we fit three models - A, B and C. Model A includes the country terms,

---

[3]https://github.com/louiskwoklf/llms-cultural-adaptability

| | persuasion | | | mobilization | | | p & m |
|---|---|---|---|---|---|---|---|
| | **human** | **GPT-3.5** | **sign agree** | **human** | **GPT-3.5** | **sign agree** | **sign agree** |
| D | +0.277 (0.009) | +0.004 (0.004) | 79% | +0.217 (0.013) | +0.018 (0.005) | 100% | 90% |
| E | +0.068 (0.028) | +0.570 (0.013) | 99% | +0.027 (0.038) | +0.142 (0.014) | 76% | 88% |
| I | -0.111 (0.028) | -0.749 (0.013) | 100% | -0.240 (0.038) | -0.907 (0.014) | 100% | 100% |
| E×I | -0.120 (0.056) | +1.111 (0.023) | 1.6% | +0.143 (0.076) | -0.170 (0.028) | 3.1% | 2.4% |
| D×E | +0.033 (0.017) | +0.040 (0.007) | 97% | +0.064 (0.024) | +0.065 (0.009) | 100% | 99% |
| D×I | +0.033 (0.017) | -0.069 (0.007) | 2.8% | +0.086 (0.024) | -0.006 (0.009) | 25% | 14% |
| D×E×I | -0.059 (0.035) | +0.010 (0.014) | 27% | -0.075 (0.047) | -0.024 (0.017) | 87% | 57% |
| | | | 58% | | | **70%** | **64%** |
| at | +0.141 (0.068) | +0.001 (0.032) | 51% | +0.208 (0.092) | -0.132 (0.034) | 1.3% | 26% |
| ch | -0.251 (0.066) | -0.011 (0.031) | 65% | +0.164 (0.090) | -0.095 (0.033) | 3.6% | 34% |
| es | +0.256 (0.075) | +0.023 (0.035) | 74% | +0.430 (0.103) | +0.137 (0.038) | 100% | 87% |
| fr | +0.435 (0.072) | +0.028 (0.034) | 80% | -0.127 (0.098) | -0.030 (0.036) | 74% | 77% |
| ge | -0.144 (0.074) | -0.085 (0.034) | 97% | +0.181 (0.101) | -0.078 (0.037) | 5.3% | 51% |
| gr | +1.111 (0.067) | +0.217 (0.031) | 100% | +0.067 (0.092) | +0.225 (0.033) | 76% | 88% |
| ie | -0.193 (0.077) | -0.096 (0.036) | 99% | +0.136 (0.105) | -0.056 (0.038) | 16% | 58% |
| il | -0.041 (0.071) | +0.103 (0.033) | 28% | -0.143 (0.098) | +0.145 (0.036) | 7.1% | 18% |
| it | +0.320 (0.077) | +0.114 (0.036) | 100% | +0.463 (0.105) | +0.176 (0.039) | 100% | 100% |
| nl | -0.178 (0.074) | -0.069 (0.035) | 97% | -0.404 (0.101) | -0.044 (0.037) | 88% | 93% |
| no | -0.213 (0.069) | -0.038 (0.032) | 88% | -0.583 (0.094) | -0.147 (0.035) | 100% | 94% |
| po | -0.483 (0.071) | +0.019 (0.033) | 29% | +0.239 (0.097) | +0.069 (0.035) | 97% | 63% |
| ro | +0.109 (0.070) | +0.004 (0.033) | 54% | +0.728 (0.095) | +0.067 (0.035) | 97% | 76% |
| se | -0.778 (0.062) | -0.041 (0.029) | 92% | -1.115 (0.085) | -0.079 (0.031) | 99% | 96% |
| uk | -0.090 (0.072) | -0.170 (0.034) | 89% | -0.242 (0.099) | -0.158 (0.036) | 99% | 94% |
| | | | **76%** | | | **64%** | **70%** |
| All | | | **70%** | | | **66%** | **68%** |

Table 1: Results from one simulation run where all prompts were in English and country of residence was stated in the prompt. The top part of the table shows the coefficients that model news framing and relative deprivation effects. This is followed by coefficients that model country-specific biases. The human and GPT-3.5 columns show the values of regression-estimated model coefficients and their standard errors. The sign agreement columns show the fraction of coefficients that agree in sign between the human- and LLM-fit models, taking account of the uncertainties in coefficient estimates. A sign agreement underlined and shown in bold indicates that the value is significantly ($p < 0.05$) greater than chance.

and the non-interaction terms for $D$, $E$ and $I$. Model B extends that by adding the two-way interaction terms $EI$, $DE$ and $DI$. Model C extends that with the three-way interaction term $DEI$. We extract coefficient estimates from the earliest of the three models in which the coefficient appears.

$$P = \bar{P} + C_i^P + \lambda_D^P D + \lambda_E^P E + \lambda_I^P I + \lambda_{EI}^P EI + \lambda_{DE}^P DE + \lambda_{DI}^P DI + \lambda_{DEI}^P DEI \qquad (1)$$

$$M = \bar{M} + C_i^M + \lambda_D^M D + \lambda_E^M E + \lambda_I^M I + \lambda_{EI}^M EI + \lambda_{DE}^M DE + \lambda_{DI}^M DI + \lambda_{DEI}^M DEI \qquad (2)$$

In total there are 44 coefficients: for each of $P$ and $M$, 15 country coefficients, plus $D$, $E$, $I$, $DE$, $DI$, $EI$ and $DEI$ coefficients (framing and relative deprivation). We compare the signs of the coefficients arising from regressing the human data with those arising from regressing LLM responses. The sign agreement for an individual coefficient takes account of the uncertainties of the estimates of the coefficient estimates, assuming normally-distributed posteriors. Sign agreements for multiple coefficients are the average of individual sign agreements. The

significance of sign agreement rates are assessed against rates for simulated data constructed by randomly shuffling the alignment between participant features and GPT-3.5 responses.

As final output of the analysis we arrange the results as shown in Table 1. The table shows sign agreement rates for several subsets of the coefficients. For the remaining analysis we focus on only two numbers: the sign agreement rate for country-specific bias terms, and the sign agreement rate for framing and relative deprivation coefficients, in both cases pooling the rates for persuasion and mobilization coefficients.

## 5 Results

### 5.1 Experiment 1: Effect of Indicating Nationality

| Coefficients | Masked | Unmasked |
|---|---|---|
| Country-specific bias terms | 51% | **70%** |
| Framing and relative deprivation coefficients | 62% | **64%** |

Table 2: Sign agreement rates for masked and unmasked nationality experiments. Agreement rates significantly ($p < 0.05$) greater than chance are underlined and shown in bold.

Table 2 compares the sign agreement rates for prompting in English with participant nationalities masked and unmasked in the pre-filled questionnaires. Sign agreement rates are significantly greater than chance only in the unmasked condition. The increase in sign agreement due to unmasking a simulated participant's nationality is much greater for country-specific coefficients (+19%) than for framing and relative deprivation coefficients (+2%).

### 5.2 Experiment 2: Effect of using a Single Language to Simulate Multinational Participants

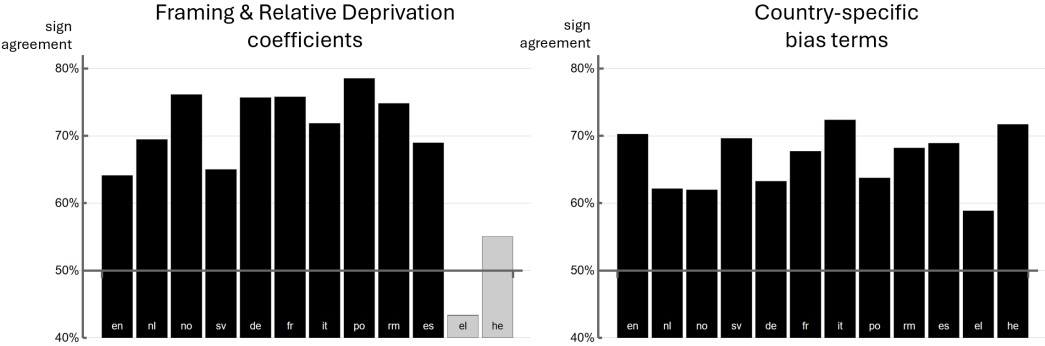

Figure 3: Sign agreement rates for monolingual prompting in 12 different languages. Agreement rates significantly ($p < 0.05$) greater than chance are shown with black bars.

Figure 3 charts sign agreement rates when prompting in different languages - in all cases, nationality is unmasked. The variability in sign agreement, depending on prompting language, is greater for framing and relative deprivation coefficients than for country-specific coefficients. In particular Greek and Hebrew prompting did not achieve sign agreement rates for framing and relative deprivation coefficients that were significantly greater than chance. While it is notable that Greek and Hebrew are the languages most lexically distant from English among the 12 languages assessed, no general trend of declining sign agreement with lexical distance was apparent.

### 5.3 Experiment 3: Effect of using Native Languages to Simulate Multinational Participants

Figure 4: Sign agreement rates for monolingual and poly-lingual prompting. Vertical lines on bars indicate +/- 1 s.d. of variation. Bars are paler when their sign agreement is not significantly ($p < 0.05$) greater than chance.

Figure 4 compares monolingual and poly-lingual prompting. For sign agreement of framing and relative deprivation coefficients no prompting scheme has a significant advantage. For country-specific coefficients, native language and country-shuffled prompting schemes perform much less well than mono-lingual prompting or full-shuffled prompting, and fail to achieve a sign agreement rate that is significantly better than chance.

## 6 Discussion

Our experiments indicate that GPT-3.5 partially replicates general psychological tendencies across cultures, despite being tasked solely with completing prompts. With a substantial sample size, GPT-3.5 mimics persuasion and mobilization effects observed in humans. Consider the prompt shown in Figure 2. It is remarkable that GPT-3.5 reproduces much of the relationship that Bos et al. (2020) demonstrated between framing and relative deprivation, and persuasion and mobilization, including the aspects that surprised Bos et al. (2020) - that anti-elitist framing increases persuasion and mobilization, while anti-immigrant framing reduces it - contrary to their predictions based on social identity that any type of out-group blaming would cause an increase.

**Experiment 1** showed that providing explicit information on simulated participants' nationalities significantly improved the fidelity of the simulation. Providing country of residence to the LLM allows it to (at least partially) replicate the international variation in the human data.

**Experiment 2** yielded unexpected outcomes contrary to initial expectations. We had anticipated trends correlating with factors such as the internet presence of a language or geographical and population-related factors. However, these anticipated trends exhibited weak correlations with the sign agreement rate per language. Instead, given nationality information, all languages performed well in terms of producing good sign agreement rates for country-specific bias terms, and for the sign agreement rates for framing and relative deprivation coefficients, only Greek and Hebrew did not manage to achieve statistical significance. It is notable that Greek and Hebrew are the only two languages that do not use letters in the Latin alphabet. In contrast, languages that used Latin characters achieved statistically significant results.

**Experiment 3** assessed the impact of native language prompting. Prompting approaches that were constant across nationalities, i.e. monolingual in experiment 2 and full-shuffled in experiment 3, performed similarly well and had statistically significant agreement rates.

On the other hand, approaches that were not constant across nationalities, i.e. native language and country-shuffled in experiment 3, performed less well and failed to achieve significance. Our interpretation is that the prompting language does influence the responses made by GPT-3.5. When the same prompting language is used for all the participants from a country these influences are sufficient to alter the fitted country coefficients. However, these alterations make the alignment with the human data worse, not better. This is the opposite of what we observed with the explicit statement of country of residence. Simply put, prompting language makes a difference, but not the correct difference.

## 7 Limitation and Future Work

Our experiments have demonstrated the capability of GPT-3.5 to utilize nationality-related information to enhance its ability to simulate human profiles. Despite the rigor of our approach as detailed in Methodology 3, a notable limitation of our study is its predominant focus on European nationals. This is a direct consequence of the constrained scope of the original study we replicated. Future research should aim to address this limitation by expanding the investigation to include a broader range of cultures and nationalities. This could be achieved through the collection of additional data or by employing new experimental setups that capture a wider spectrum of cultural backgrounds.

Moreover, this study primarily examines the performance of a single model, GPT-3.5. We posit that there is significant potential in extending this research to develop a comprehensive benchmark that assesses cultural adaptability across various models.

## 8 Conclusion

In this study, we investigate how conveying the cultural background of a human profile through nationality-related information, either by specifying country of residence or by native language prompting, can affect GPT-3.5's ability to simulate such a profile. We also examine how the representation of national traits varies across different multilingual settings.

To do so, we replicate a survey-styled psychological experiment, analyzing reactions to persuasive and mobilizing stimuli by having GPT-3.5 simulate responses of 7,286 participants from 15 countries. Our results show that explicitly stating country of residence improves simulation fidelity of a human profile, allowing national variations in persuasion and mobilization to be modeled. However, conveying additional nationality traits with native language prompting decreases alignment with human responses, with certain languages having a particularly negative impact on the results. For nationality-specific modeling, it causes response changes that do not align with the international variation in humans. For the effects of framing and relative deprivation on persuasion and mobilization, which are the primary concern of the original study by Bos et al. (2020) that we simulate, prompting in Greek or Hebrew greatly reduced the fidelity of the simulation.

Overall, our findings suggest two key insights: (i) explicitly stating the nationality of a human profile enhances GPT-3.5's effectiveness at simulating their behavior, and (ii) simulating participants in their native language negatively affects the simulation's fidelity, with Greek and Hebrew being particularly detrimental in our experimental setup.

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

# A   Appendix

## A.1   Majority Languages and Number of Participants of Countries

| Country | Majority Language | ISO 639 Language Code |
|---|---|---|
| Netherlands | Dutch | NL |
| Ireland
United Kingdom | English | EN |
| France | French | FR |
| Austria
Germany
Switzerland | German | DE |
| Greece | Greek | EL |
| Israel | Hebrew | IW |
| Italy | Italian | IT |
| Norway | Norwegian | NO |
| Poland | Polish | PL |
| Romania | Romanian | RO |
| Spain | Spanish | ES |
| Sweden | Swedish | SV |

Table 3: Countries with their primary languages and corresponding ISO codes.

| Country | Number of Participants |
|---|---|
| Austria | 529 |
| France | 528 |
| Germany | 414 |
| Greece | 548 |
| Ireland | 384 |
| Israel | 461 |
| Italy | 446 |
| Netherlands | 377 |
| Norway | 433 |
| Poland | 549 |
| Romania | 659 |
| Spain | 469 |
| Sweden | 519 |
| Switzerland | 512 |
| United Kingdom | 458 |

Table 4: Number of simulated participants per country, totaling 7286 participants.

## A.2    News Article Templates used in Simulations

---

## INSTRUCTION – Please read the following online news article closely.

Title: Purchasing power will decline - foundation FutureNow releases new report

Photo: <description – hands and open empty wallet, owner is looking within>

Text: According to a new report by FutureNow purchasing power will decline in the coming years. A spokesperson for the independent foundation that has been monitoring economic developments for years comments on the report: 'We have to raise awareness about what this prospect means. There will be less money to spend. Action has to be taken now to address this threat.'

---

Figure 5: News article template without populist framing.

---

## INSTRUCTION – Please read the following online news article closely.

Title: Purchasing power will decline for [nationals] - foundation FutureNow blames politicians in new report

Photo: <description – hands and open empty wallet, owner is looking within>

Text: According to a new report by FutureNow purchasing power in [country] will decline in the coming years. A spokesperson for the independent foundation that has been monitoring economic developments for years comments on the report: 'The common citizens in [country] need to be made aware of the fact that they will have less money to spend. So many people in [country] are working so hard everyday to have a good life. There is something profoundly wrong when these efforts do not pay off. It is obvious that immigrants are to blame. They are too demanding, they exploit our system and are hard to integrate. Action has to be taken now to address this threat to the well-being of our people.'

---

Figure 6: News article template with anti-elite framing. [nationals] and [country] are placeholders to be substituted depending on the nationality of the simulated participant. In masked-nationality simulations, they were removed from the prompt.

---

## INSTRUCTION – Please read the following online news article closely.

Title: Purchasing power will decline for [nationals] - foundation FutureNow blames politicians in new report

Photo: <description – hands and open empty wallet, owner is looking within>

Text: According to a new report by FutureNow purchasing power in [country] will decline in the coming years. A spokesperson for the independent foundation that has been monitoring economic developments for years comments on the report: 'The common citizens in [country] need to be made aware of the fact that they will have less money to spend. So many people in [country] are working so hard everyday to have a good life. There is something profoundly wrong when these efforts do not pay off. It is obvious that politicians are to blame. They have been too short-sighted, self-serving, and corrupt in recent years. They don't care about anyone but themselves and are too detached from the people. Action has to be taken now to address this threat to the well-being of our people.'

---

Figure 7: News article template with anti-immigrant framing. [nationals] and [country] are placeholders to be substituted depending on the nationality of the simulated participant. In masked-nationality simulations, they were removed from the prompt.

## INSTRUCTION – Please read the following online news article closely.

Title: Purchasing power will decline for [nationals] - foundation FutureNow blames politicians in new report

Photo: <description – hands and open empty wallet, owner is looking within>

Text: According to a new report by FutureNow purchasing power in [country] will decline in the coming years. A spokesperson for the independent foundation that has been monitoring economic developments for years comments on the report: 'The common citizens in [country] need to be made aware of the fact that they will have less money to spend. So many people in [country] are working so hard everyday to have a good life. There is something profoundly wrong when these efforts do not pay off. It is obvious that politicians and immigrants are to blame. Politicians have been too short-sighted, self-serving, and corrupt in recent years. Migrants are too demanding, they exploit our system and are hard to integrate. And still, politicians only take care of the migrants instead of our own people. Action has to be taken now to address this threat to the well-being of our people.'

Figure 8: News article template with both anti-elite and anti-immigrant framing. [nationals] and [country] are placeholders to be substituted depending on the nationality of the simulated participant. In masked-nationality simulations, they were removed from the prompt.

