# OpenReview forum: "Evaluating Cultural Adaptability of a Large Language Model via Simulation of Synthetic Personas"
_colmweb.org/COLM/2024/Conference — COLM_

### Official Review · Reviewer_aYm5 · 2024-04-19

**Rating:** 9
**Confidence:** 5
**Ethics Flag:** 1

**Summary:**

Simulating human responses to a study from psychology using GPT-3.5 prompted with respondents demographic information, this study shows high fidelity, which is improved by also indicating the respondent's country, but deteriorates when using the respondent's language for prompting. Best fidelity is with English prompts, with lexical distance from English being negatively correlated with fidelity for other languages.

**Questions To Authors:**

Which language did participants in Bos et al.'s study were exposed to? What story did they read? The same as in the present study? What country was the story about? Their own? Given these details, can you speculate on why the prompting language yielded the opposite effect than that expected in experiment 3?

How was the country masked in the present study?

What is the justification to compare the sign of the country coefficients? Since they're set to have zero mean, the sign just means where each country lies relative to the mean of the countries in this specific set of countries, and may be very different if a different set of countries is selected.

Figure 3 should be a table.

P values should be given in decimals and not percents.

Greek and Hebrew are not language families.

**Reasons To Accept:**

Addresses a topic of interest to a wide audience and societal impact, of cultural sensitivity of GPT.

Grounded in psychological research and using data from robust experiments with human participants as a basis for comparison.

Results are surprising and impactful as they reveal the capability of simulating human tendencies but confirm cultural biases in LLMs, and as simulating human responses with high fidelity across cultures is a useful task in itself.

**Reasons To Reject:**

Lacking details about the experimental setup of this study as well as crucial details from the Bos et al. study. (see questions to authors). The authors follow the analysis "closely, but not exactly", but it's not clear how they're different.

Missing details about geographical, population-related and linguistic factors in the analysis of experiment 2, and about the correlation obtained with them.

Testing the validity of translation only with GPT-3.5 is insufficient (we don't know how reliable this is without correlating it with human judgments) and it's also unclear how exactly this is done.

Missing context and citations for studies that also investigated the effect of prompting country/language on LLM fidelity in simulating human responses:
https://arxiv.org/abs/2306.16388
https://aclanthology.org/2023.c3nlp-1.7/
https://aclanthology.org/2023.c3nlp-1.12/

Averaging the relative deprivation ratings and replacing the original ratings seems arbitrary and unjustified. It is fair that it improves fidelity but what happens without doing this?

The results from experiment 3 de presented in figure 4 with seemingly arbitrary x value, which is confusing. If there's no meaning to the x value it should be in a separate subfigure.

Framing the results as revealing human-like characteristics, and claiming "LLMs exhibit persuasion and mobilization effects akin to humans", is unnecessary anthropomorphism. They are only simulators. Also, "LLMs" is to general since only GPT-3.5 was studied.

---

> ### Author Rebuttal · Authors · 2024-05-31
>
> **Bos's analysis followed ‘closely but not exactly’. Aren’t country coefficients relative to the mean of this specific set of countries?**
>
> Bos used Poland as a reference, forcing its coefficient to 0. We instead express country coefficients relative to a zero-mean, allowing comparison for all countries. Since the country set is fixed, we do not see an issue with this.
>
> **Missing correlations**
>
> Fidelity vs. internet presence (+0.22); fidelity vs. speaker population (-0.19).
>
> **Limitations of machine translation**
>
> We believe that the ‘google translations’ we used are sense-preserving, but we accept that lack of human validation is a fair criticism. We are compiling validated translations and our final version will present data based on these. As an interim measure, we have re-run our simulation using superior GPT-4 translations. Our conclusions are unchanged in all important aspects.
>
> **Averaging of Relative Deprivation (RD)**
>
> This was a compromise to our modeling that got baked in that we thought could help improve model fidelity. We have re-simulated without this hack, and our conclusions are unchanged. The final version will reflect this new data.
>
> **Arbitrary x-value for exp-3 results in fig-4**
>
> The x-value is the average distance-from-English of the prompting languages used in exp-3. We will fix the omission.
>
> **Unnecessary anthropomorphism**
>
> We will revise “exhibit effects akin” to “mimic the effect”.
>
> **Cannot generalize to LLMs since only GPT-3.5 was studied**
>
> We never meant to make this claim, and will correct our wording to make this clear.
>
> **Bos study: language, story, country. Why differences in experiment 3**
>
> Each Bos participant completed a questionnaire in their native tongue, featuring a news story about their own country. In section 6 we conclude that GPT-3.5 does respond differently depending on the prompting language, but those differences are different from those seen in human participants. We are hesitant to speculate on the cause without sufficient evidence.
>
> **Method to mask nationality**
>
> Nationality answers were omitted from the prompt. Within the article, ‘France’ (for example) was replaced by ‘the country’ or ‘our country’ and ‘French citizens’ by ‘nationals’. We will explain in our final version.
>
> **Other mentioned issues**
>
> Thanks, we will correct them in our final version.

---

> > ### Comment · Reviewer_aYm5 · 2024-05-31
> >
> > Thank you for the clarifications. I raised my score to 9.

---

### Official Review · Reviewer_KEfE · 2024-05-11

**Rating:** 8
**Confidence:** 3
**Ethics Flag:** 1

**Summary:**

The goal of this paper is addressing the research question “How to effectively incorporate language in LLMs?” as well as “How precise is the nationality trait depicted by multilingual LLMs?” towards personalized and inclusive LLMs. Here, a semi-complete questionnaire (adapted from Bos study) is provided where numbers should be selected for each question along with news stimuli to the participants. The study contains >7K participants across 15 countries.

3 GPT-3.5-Turbo-1106 personas are created : 1) directly expressing the nationality 2) use of a side language (majority language of each country) 3) use of native language – Following each experiments, each participant’s persuasion and mobilization scores were collected while simulating the participant’s reaction to news stimuli.

While modern LLMs have multilingual capabilities, there is bias due to shortage of data in other languages than English ( or other similar high-resource languages), this is a challenging problem specially when it comes to low-resource languages. Further, there are gender biases in certain languages that can be also present in English LLMs. Also, in tasks such as emotion detection, English LLMs perform better than other languages.

LLM persona was introduced in 2023 where LLMs can showcase certain personality traits when interacting with an end-user. It is also shown that RLHF produces more liberal content than the original pretrained data that is more conservative.

 Experiment result was that languages that resemble English language the most have the higher agreement score and that translating to native language yields poor results. Further, providing the nationality of the participant as part of the prompt provides better results. Based on the experiments, better results are yielded when English is chosen as the language of prompting.

**Questions To Authors:**

I am not sure if using Google Translate would be best resource in this scenario since even in case of low-resource languages, Google Translate doesn't have the best performance. Could you please explain how you handled this situation?

Also could you please elaborate a bit on if you handled gender biases present in certain nationalities? or what would you propose in this regard?

**Reasons To Accept:**

- Authors have performed tests to ensure linguistic meaning of the survey was not lost during translation in different languages.

- Comprehensive experiments across a diverse set of countries and a huge number of participants followed by detailed discussions.

- This research paper provides insight in design multilingual LLMs that needs to be more inclusive.

- The multidisciplinary nature of this work is beneficial to the community.

**Reasons To Reject:**

I am pro accepting this paper specially due to its multidisciplinary effect.

---

> ### Author Rebuttal · Authors · 2024-05-31
>
> Thank you for your positive and insightful feedback. We are encouraged that you find our multidisciplinary work beneficial to the community, especially in designing more inclusive multilingual LLMs. We are glad that you found our experiments comprehensive across a diverse and large scale. By simulating over 7,000 human responses given their profiles, we carried out three experiments to test: 1) the effect of explicitly mentioning nationality, 2) the effect across various countries, and 3) the effect of shuffling pairings of the country and its native language. We appreciate your acknowledgment of our testing efforts to preserve linguistic meaning in our prompt translations. We would like to address your questions and concerns below and incorporate your feedback in our final version.
>
> **Limitations of machine translation**
>
> We believe that the ‘google translations’ we used are sense-preserving, but we accept that lack of human validation is a fair criticism. We are compiling validated translations and our final version will present data based on these. As an interim measure, we have re-run our simulation using superior GPT-4 translations. Our conclusions are unchanged in all important aspects.
>
> **Considering gender biases**
>
> We have considered potential gender biases in two ways:
> 1. Some languages may have gendered words. Bilingual speakers have checked our translations and have confirmed that the gendering follows standard translation patterns and does not include any contentious choices such as translating ‘the doctor’ to a masculine form.
> 2. Following Bos et al., our prompts do explicitly state the gender of the simulated subject, and nothing else in the prompt refers to or indicates gender. Neither Bos nor ourselves have tested whether there are gender differences in the human data. Assessing whether LLM simulation reproduces any gender differences if they exist, or introduces new false ones, are interesting topics that we hope to pursue in the future.

---

> > ### Comment · Reviewer_KEfE · 2024-06-07
> > **Increasing my rating**
> >
> > After reading the authors' detailed responses to reviewers, I am changing my rating to 8. I suggest containing some of these concerns/questions and responses in paper or its appendix or an accompanying website since if this is raised by us, it could be someone's else question too.

---

### Official Review · Reviewer_sKd3 · 2024-05-12

**Rating:** 7
**Confidence:** 4
**Ethics Flag:** 1

**Summary:**

This paper replicates a psychological experiment from Bos et al. (2020), where predefined human profiles are simulated to study their reactions to news stimuli. The authors have access to the original human responses of Bos et al. (2020)'s experiment, and prompt GPT3.5 to generate simulated profiles. The authors test various combinations of information given to the LLM during prompting, but the main goal of the study is to explore how the various "metadata" information can affect the output of the LLM. I refer to it as "metadata" even if it is not, it is more a set of additional information about the human profile: nationality and language.
Through a set of analysis and combinations of adding information about nationality, language, both, or neither, the authors explore how this affects the final prediction of the LLM.
As far as I understand it, the final prediction of the LLM is a simple numerical value as response to one question precedes by a longer prompt presenting the human profile, the relative deprivation values of the human, as well as a news article.
The paper is well written and interesting, but I have some doubts about some aspects of the motivation and claims. Please see weaknesses and questions for details.

**Questions To Authors:**

- In experiment 2, I am not sure I understand the sentence "the simulation was run 12 times, each time simulating in the same language regardless of the nationality of the simulated participant". What is the single language used? Is this English?

- In experiment 3: I am not sure I understand why you have approached this experiment in this way. If we want to test the effect of the language, shouldn't the prompt be change to the relative language of each country? Or is this to test if English yields the best results overall? I am not sure about the motivation here.

- Did I get it correct that your experiments are based on one numerical value returned by GPT3.5?

**Reasons To Accept:**

- A nice contribution in the direction on grounding interdisciplinary research by replicating a study from psychology on LLMs.

- The authors present a number of experiments and analysis of their results.

**Reasons To Reject:**

- Some aspects of the paper are not clear to me, and the authors fail to give clear arguments to support some of their decisions (I am not sure if I understand some of the decisions made by the authors, please see questions).

- While I think that the idea of the work is interesting. Some I am not sure if asking GPT3.5 to give one numerical value means that we have a system that effectively simulates human behavior.

- During their experimentation, the authors relied on machine translation to translate their prompts. To assess the correctness of their translation they used GPT3.5. While this might work and yield good results, it does not equate human evaluation. The authors fail to discuss this aspect in their limitations section, nor discuss the effect this can have on languages with higher lexical distant to English (which is one of their findings, that for languages with a high lexical distant from English, using English prompts is better). Maybe this is mainly a side effect of bad translation? It is anyways not discussed by the authors, and I did miss this discussion.

---

> ### Author Rebuttal · Authors · 2024-05-31
>
> Thank you for your insightful feedback. We are encouraged that you find our work to be a nice contribution for grounding interdisciplinary research. We appreciate your acknowledgement of our presenting numerous experiments and analyses, and we are glad that you find the idea of our work interesting.
>
> **Just one numerical response is not enough to claim simulation of human behavior.**
>
> Not one. As described in section 3.5, we followed the Bos et al. study. Subjects (human and LLM) rated agreement with three statements to assess persuasion, and two to assess mobilization; each set of ratings was averaged to give two responses per subject. We make no claim to a general simulation of human behavior, just to the narrow but interesting aspect tested by the Bos et al. study which has generated three publications with ~300 citations between them.
>
> **Limitations of machine translation**
>
> We believe that the ‘google translations’ we used are sense-preserving, but we accept that lack of human validation is a fair criticism. We are compiling validated translations and our final version will present data based on these. As an interim measure, we have re-run our simulation using superior GPT-4 translations. Our conclusions are unchanged in all important aspects.
>
> **Language used in exp-2?**
>
> We wrote:
> “In the second experiment…the language of the prompt was varied. The simulation was run 12 times, each time simulating in the same language regardless of the nationality of the simulated participants. The languages used are the majority-spoken language of each country as shown in Table 3 in the Appendix.”
>
> To be clearer we will insert:
> “On the first run all subjects, regardless of nationality, were prompted in English. On the second run all in French. On the third German. Etc."
>
> **Language used in exp-3?**
>
> In exp-3 (main condition) subjects were indeed prompted in their native languages i.e. French subjects in French, German in German, etc. In the country-shuffled condition the pairing of nationality and language was randomly permuted, so (for example) Germans in French, Poles in Spanish, etc. In the fully-shuffled condition the pairing of subjects and language was randomly permuted, so some English were prompted in French, some in Polish, etc. and similarly for other nationalities.

---

### Official Review · Reviewer_mnFL · 2024-05-14

**Rating:** 6
**Confidence:** 5
**Ethics Flag:** 1

**Summary:**

The paper examines how information about language and the country of residence influences the replication of a multi-country experiment on political persuasion and mobilization.
To this end, the paper relies on data from a political experiment involving ~7k participants from 15 European countries who read prompts about financial issues in their countries. The prompts attributed blame for the issues to either elites, immigrants, or both, and participants' persuasion and mobilization levels were assessed.
The current research replicates this study, adding participants' demographic information and using an LLM to predict responses. Three experimental settings are tested where the prompts vary by including individuals' country of residence and modifying the language to compare the LLM's performance across conditions.

**Questions To Authors:**

The evaluation method assesses 44 coefficients, 30 of which are country-level (C_i^P, C_i^M). However, the LLM is evaluated on replicating these coefficients even though the country of residence is not provided in the prompt. Could you clarify whether the expectation is for LLMs to infer this information from the language and text of the prompt, and if so, what linguistic cues or patterns the LLM might be utilizing for this inference?

The relative deprivation appears to be a crucial independent variable in Bos et al.'s experiment. However, your paper mentions that "the three relative deprivation ratings in our prompt were set to the same average of the actual three relative deprivation ratings by the human participant in the original study" to enhance fidelity. Does this adjustment imply that LLMs struggle to capture the association between relative deprivation and persuasion (and mobilization), and if so, what are the potential reasons for this limitation and how does that impact the paper's main goal for replicating the human responses?

**Reasons To Accept:**

Incorporating and examining cultural information embedded in LLMs is a crucial task, our field increasingly needs accurate approaches for capturing cultural representations and the role of language in shaping them. The paper has this valuable motivation.

This paper makes a valuable point by investigating the correlation between various linguistic factors (e.g., distance from English) and cultural representation, exploring whether and how language influences cultural representation in LLMs.

The research design is noteworthy for its reliance on human data from diverse countries, adding validity to the study of this important question.

**Reasons To Reject:**

The paper's approach to assessing the fidelity of LLMs in simulating human responses raises concerns. The chosen method, focusing on replicating regression results, seems inadequately justified. A simpler correlation analysis might have provided a more direct measure of how well the LLMs simulate individual responses.

Additionally, the paper omits mentioning that the data and experiments are limited to European countries and languages. While this doesn't necessarily invalidate the findings, it's a surprising omission that should have been acknowledged.

The generalization of findings to psychological traits is also problematic, as the dataset specifically examines the impact of two contexts on agreement with political arguments about a financial issue. A more appropriate dataset would have included psychological traits collected across diverse cultures, such as those found in the World Values Survey. Furthermore, while the paper discusses culture, it neglects to address the limited cultural variance within European countries.

---

> ### Author Rebuttal · Authors · 2024-05-31
>
> **Correlation better than sign agreement**
>
> Correlation would be a powerful measure if our aim was to predict the dependent variables of persuasion and mobilization from the independent subject and framing variables, but for our aim of simulating the dependencies between the independent and dependent variables it is a blunt tool.
>
> For example, consider samples from N(0,1)^100. Let a dependent variable be linearly dependent on those variables with coefficients of {-1,1} for the first 99, and +10 for the 100th. Suppose a simulation of the dependent variable had sufficient fidelity that it reproduced a linear relationship with the exact same coefficients, except that the 100th was -10 rather than +10. The correlation between the simulated and real data would be ~0.0, whereas the sign agreement would be 99%.
>
> **Failure to characterize the countries**
>
> This was an inept omission. In our final version, we will clarify that European (plus Israel) countries and languages are used, and note this limitation.
>
> **Failure to model values and cultural variance**
>
> Representation of these factors within LLMs is interesting and important. The Bos study has generated three publications with 300 citations between them showing that the relationships it investigates are of interest even without incorporation of these additional factors.
>
> **Modeling nationality when masked**
>
> In exp-1 we compared nationality being masked vs stated - simulation fidelity increased from 59% to 71%. Based on this, in exp-2 and exp-3 nationality was stated. In exp-1, when the nationality was masked and all prompting was in English, the LLM does indeed have no cue to nationality and so gets the correct sign for the 30 country coefficients only randomly. It is still able to model the relationship between framing and relative deprivation variables, and ratings of persuasion and mobilization (the other 14 coefficients).
>
> **Averaging of Relative Deprivation (RD)**
>
> This was a compromise to our modeling that got baked in that we thought could help improve model fidelity. We have re-simulated without this hack, and our conclusions are unchanged. The final version will reflect this new data.

---

> > ### Comment · Reviewer_mnFL · 2024-06-06
> >
> > Thanks for your response,
> >
> > With regard to the correlation test, I would still suggest a normalized way to test the correlation. I still believe that comparing the coefficient signs is not a robust test.
> >
> > While I acknowledge the relevance of Bos' research, I also believe that your introduction and research questions explore a broader scope beyond the specific context of their experiment. You can refine these sections to ensure they are more focused and directly address the key findings of your study, making your claims clearer and more concise.

---

> > > ### Author Response · Authors · 2024-06-07
> > >
> > > Thank you for your feedback on our rebuttal.
> > >
> > > Regarding a correlation test, we appreciate your persistence in suggesting this. We have disentangled the independent variables and have applied a correlation test to country-related coefficients (which are commensurate), while using our sign agreement test for non-country-related coefficients (which are not). We are pleased to find that the results have become clearer, while our conclusions remain unchanged in all important aspects. We will include our latest findings in the final version.
> > >
> > > We also appreciate your acknowledgment of the relevance of Bos’ research alongside ours. We understand your point about the scope of our introduction and research questions. We will refine these sections to ensure they directly address our key findings in the final version, particularly by limiting generalizations of traits and culture to the effects of persuasion and mobilization.
> > >
> > > Thank you once again for your valuable insights.

---

### Decision · Program_Chairs · 2024-07-10

**Decision:**

Accept

**Comment:**

The paper explores the ability of a language model to understand and replicate human profiles from different nationalities. The work is novel, interdisciplinary, scalable, and addresses topics of rising interest to the community, such as pluralism in AI.
The authors are asked to include more clarity re the following in the camera ready.
1. Limitations of their approach, such as the use of unverified automatic translations, and that their study only covers European nations who might be more closely aligned in values and perspectives as compared to other diverse nations worldwide.
2. Choice of framework/study for moral values: Bos versus World Values Survey. Each have their pros and cons and need to be disucssed in the paper, rather than setting up an arbitrary choice.
3. Clarity of experimental setup, e.g., use of one numerical values returned by the model, languages used in the various experiments.

The authors are encouraged to overall increase the clarity in writing of their paper and explicitly state limitations. They are also asked to incorporate reviewer feedback into their final version.

[comments from PCs]
Technical aspects aside. This paper on the one hand says it’s focused on culturally aware LLM, which is interesting. But title is really focused on simulating human surveys using LLMs, which is the experiment they ran. The intro suggests this is the experimental methodology only, but this is not what the title says. Therefore, we observe a discrepancy in the methods used versus claims. It may be the case that the ‘embedding’ in the title is interpreted differently by authors. **We ask the authors to fix this contextualization and presentation issue, including, if needed, by modifying the title.** Please consider how the audience in the conference will interpret your title.